

# Comprehensive Atmospheric Modeling of Reactive Cyclic Siloxanes and Their Oxidation Products

Nathan J. Janechek[1,2], Kaj M. Hansen[3], and Charles O. Stanier[1,2]

[1]Department of Chemical and Biochemical Engineering, University of Iowa, Iowa City, IA 52242, USA
[2]IIHR Hydroscience and Engineering, University of Iowa, Iowa City, IA 52242, USA
[3]Department of Environmental Science, Aarhus University, Roskilde, Denmark

*Correspondence to*: Charles O. Stanier (charles-stanier@uiowa.edu)

**Abstract.** Cyclic volatile methyl siloxanes (cVMS) are important components in personal care products that transport and react in the atmosphere. Octamethylcyclotetrasiloxane ($D_4$), decamethylcyclopentasiloxane ($D_5$),
dodecamethylcyclohexasiloxane ($D_6$), and their gas phase oxidation products have been incorporated into the Community Multiscale Air Quality (CMAQ) model. Gas phase oxidation products, as the precursor to secondary organic aerosol from this compound class, were included to quantify the maximum potential for aerosol formation from gas phase reactions with OH. Four 1-month periods were modeled to quantify typical concentrations, seasonal variability, spatial patterns, and vertical profiles. Typical model concentrations showed parent compounds were highly dependent on population density as cities had
monthly averaged peak $D_5$ concentrations up to 432 ng m$^{-3}$. Peak oxidized $D_5$ concentrations were significantly less, up to 9 ng m$^{-3}$ and were located downwind of major urban areas. Model results were compared to available measurements and previous simulation results. Seasonal variation was analyzed and differences in seasonal influences were observed between urban and rural locations. Parent compound concentrations in urban and peri-urban locations were sensitive to transport factors, while parent compounds in rural areas and oxidized product concentrations were influenced by large-scale seasonal
variability in OH.

## 1 Introduction

Cyclic volatile methyl siloxanes (cVMS) are present in a wide range of personal care and cosmetic products (e.g. hair products, lotions, antiperspirants, makeup, and sunscreens) as well as in sealers, cleaning products, and silicone products (Wang et al., 2009; Horii and Kannan, 2008; Dudzina et al., 2014; Lu et al., 2011; Capela et al., 2016). As high production volume chemicals
(>4.5x10$^5$ kg yr$^{-1}$ produced or imported to the U.S.) their environmental fate is an important topic. The most prevalent cVMS species in personal care products is decamethylcyclopentasiloxane ($D_5$), although octamethylcyclotetrasiloxane ($D_4$) and dodecamethylcyclohexasiloxane ($D_6$) are also emitted (Horii and Kannan, 2008; Dudzina et al., 2014; Wang et al., 2009; Lu et al., 2011). Atmospheric lifetimes (Atkinson, 1991) are approximately 5-10 days at typical OH concentrations; accordingly, long-range transport (Xu and Wania, 2013; Krogseth et al., 2013a; McLachlan et al., 2010; Genualdi et al., 2011; MacLeod et
al., 2011) of cVMS occurs. The environmental fate and transport of cVMS has been widely studied due to concerns of





bioaccumulation and persistence in the environment (Wang et al., 2013; Rucker and Kummerer, 2015). Several regulatory screenings in Canada (Environment Canada and Health Canada, 2008a, b, c), U.K. (Brooke et al., 2009a, b, c), Netherlands (Smit et al., 2012), and Nordic countries (Kaj et al., 2005a; Kaj et al., 2005b; Lassen et al., 2005) have studied the environmental impact of parent cVMS, and while cVMS is widespread, it is not expected to pose a risk to the environment at observed concentrations (Siloxane D5 Board of Review, 2011; Fairbrother et al., 2015; Gobas et al., 2015; Fairbrother and Woodburn, 2016). However, some debate still exists as the European Chemicals Agency recently proposed $D_4$ and $D_5$ restrictions in wash-off (e.g. shampoo) personal care products due to concerns on the aquatic environment (ECHA, 2015). Fate and transport of cVMS is summarized as emission (mainly to the atmosphere) in population centers as a result of personal care product (PCP) use (Mackay et al., 2015; Montemayor et al., 2013; Gouin et al., 2013), followed by atmospheric transport and reaction by the hydroxyl radical (OH) (Xu and Wania, 2013). Emissions and concentrations are highly dependent on population, with urban locations (Yucuis et al., 2013; Genualdi et al., 2011; Krogseth et al., 2013b; Buser et al., 2013a; Companioni-Damas et al., 2014; Ahrens et al., 2014) and indoor environments (Tang et al., 2015; Yucuis et al., 2013; Companioni-Damas et al., 2014; Pieri et al., 2013; Tri Manh and Kannan, 2015) having much higher concentrations than remote locations.

Substantial insights regarding cVMS fate, transport, and expected concentrations have come from atmospheric modeling studies. McLachlan et al. (2010) simulated atmospheric $D_5$ concentrations using a hemispheric scale 3D atmospheric chemistry and transport model (McLachlan et al., 2010; Genualdi et al., 2011; Krogseth et al., 2013a). MacLeod et al. (2011) simulated $D_5$ globally using a multimedia mass balance at 15° horizontal resolution (Genualdi et al., 2011; MacLeod et al., 2011). Global zonally averaged modeling using the multimedia GloboPop model has also been performed (Xu and Wania, 2013; Wania, 2003). Emission estimates have been back-calculated from measured atmospheric concentrations using a multimedia model (Buser et al., 2013a; Buser et al., 2014), and compartmental model studies focusing on specific partitioning or loss processes have also been conducted (Navea et al., 2011; Whelan et al., 2004). These modeling studies have permitted extension, both in time and space, beyond the sparse measurement dataset and testing of key model processes (emissions, fate, and transport) versus modeled concentrations. Latitudinal gradients, urban-rural-remote gradients, seasonal patterns, sensitivity to processes and parameterizations, and diel cycles have been explored using these models. For example, typical concentrations of $D_5$ in well-mixed air in urban locations are thought to be in excess of 50 ng m$^{-3}$, while remote concentrations may exhibit $D_5$ concentrations from 0.04 to 9 ng m$^{-3}$ (Navea et al., 2011; Krogseth et al., 2013a).

Atmospheric measurements of cyclic siloxanes have been performed in ambient air (McLachlan et al., 2010; Genualdi et al., 2011; Yucuis et al., 2013; Ahrens et al., 2014; Kierkegaard and McLachlan, 2013; Krogseth et al., 2013b; Krogseth et al., 2013a; Buser et al., 2013a; Companioni-Damas et al., 2014). Higher concentration microenvironments have also been surveyed through measurement (WWTP, landfills, and indoor air) (Krogseth et al., 2013b; Cheng et al., 2011; Wang et al., 2001; Pieri et al., 2013; Yucuis et al., 2013; Tri Manh and Kannan, 2015; Companioni-Damas et al., 2014; Tang et al., 2015). In several instances, model-measurement comparison has been conducted and to a large extent, confirmed our understanding of emissions, fate and transport. Generally good agreement for rural and remote locations have been observed (McLachlan et



al., 2010; Krogseth et al., 2013a; MacLeod et al., 2011; Navea et al., 2011; Xu and Wania, 2013; Genualdi et al., 2011) while urban areas tend to be under predicted (Genualdi et al., 2011; Yucuis et al., 2013; Navea et al., 2011). Measured seasonal concentration variations have been replicated for sites in rural Sweden and the remote Arctic. However, it was noted that the DEHM model tended to have better agreement during summer/fall compared to winter (McLachlan et al., 2010; Krogseth et al., 2013a). The BETR model conversely had better agreement during winter compared to summer for the same rural Sweden site (MacLeod et al., 2011).

The majority of modeling and chamber study investigation, and all of the ambient measurements for cVMS, have focused on the emitted or "parent" cVMS compounds (i.e., $D_4$, $D_5$, and $D_6$). The identity and fate of the cVMS oxidation products has received less scrutiny until recently, compared to the parent compounds. Sommerlade et al. (1993) reacted $D_4$ with OH in an environmental chamber and identified multiple reaction products by GC-MS, with the single OH substituted silanol ($D_3TOH$) as the most prevalent resolved species, with species identification confirmed by matching retention time and mass spectra compared to synthesized $D_3TOH$ (Sommerlade et al., 1993). Because of the method of collection (the product was collected from rinsing the environmental chamber walls with solvent) confirmation of secondary aerosol production from $D_4$ oxidation was not possible from Sommerlade et al. (1993). Chandramouli and Kamens (2001) reacted $D_5$ in a smog chamber, with separate analysis of gas and aerosol products, confirming the presence of $D_4TOH$ in the GS/MS analysis of the condensed aerosol phase.

Wu and Johnston (2016) conducted more exhaustive characterization of aerosols from photooxidation of $D_5$, using high performance mass spectrometry, revealing both monomeric and dimeric oxidation products, with molar masses up to 870. Oxidation progressed not only by substitution of a methyl group with OH (e.g. leading to $D_4TOH$), but also by substitution with $CH_2OH$; linkages between Si-O rings to form dimers were through O, $CH_2$, and $CH_2CH_2$ linkage groups.

Aerosols containing Si and likely from photooxidation of gaseous precursors have been recently identified in multiple locations in the U.S. using laser ablation particle mass spectrometry of ultrafine particles (Bzdek et al., 2014). Bzdek et al. (2014) contend that a photooxidation source is most consistent with observations because of the times of day of occurrence, short atmospheric lifetime of the particle size in question (10-30 nm), lack of wind direction dependence that would be expected from primary sources, ubiquity across disparate measurement sites, and similarity in temporal evolution of nanoaerosol Si to other species with known photochemical sources. Except for the reports of the concentrations of ambient oxidized cVMS in Bzdek et al. (2014), there are no ambient measurements or model-based estimates of the potential aerosol concentrations from cVMS oxidation. This work begins to address that gap by simulating the gas phase oxidation product concentrations using the atmospheric chemistry and transport model Community Multiscale Air Quality (CMAQ). As experimental determinations of aerosol yield become available, the simulations can be updated to include secondary organosilicon aerosol concentrations.

This work builds on the limited information available on the oxidation products. Properties relevant to fate and transport (e.g. Henry's law coefficient) have been predicted in this work and in others based on structure activity relationships (Buser et al., 2013b; Whelan et al., 2004). Latimer et al. (1998) measured equilibrium gas particle partitioning of $D_5$ and $D_4TOH$ on diesel, wood, coal soot, and Arizona fine dust aerosols. Whelan et al. (2004) performed equilibrium air-particle



and air-cloud droplet partitioning modeling of multiple substituted OH silanols. More extensive information is available about the gas-particle partitioning (Latimer et al., 1998; Tri Manh and Kannan, 2015; Tri Manh et al., 2015; Kim and Xu, 2016) and aerosol phase reactions (Navea et al., 2011; Navea et al., 2009a; Navea et al., 2009b) of the precursor compounds, but these confirm that the gas-phase oxidation and transport of the parent compounds are substantially more important than the heterogeneous oxidation pathways and thermodynamic partitioning of the parent compounds onto ambient aerosols.

In this work, atmospheric gas phase concentrations of $D_4$, $D_5$, $D_6$, and its oxidization products are modeled comprehensively using the chemical transport CMAQ model. The purpose of the model-based investigation is twofold. First, it enables the highest resolution (36 km) to-date simulation of the parent compound over the U.S.; the model simulates vertical profiles, urban-to-rural transitions, and the dependence of these on factors such as season and mixed layer height. Second, this paper reports, for the first time in detail, concentrations of the cVMS oxidation products. Some fraction of products is likely distributed into the aerosol phase, thus contributing to aerosol Si concentrations on regional and global scales. We expand upon the modeling first presented in Bzdek et al. (2014), but with improved emission estimates, inclusion of wet and dry deposition, and incorporation of season-dependent boundary conditions.

## 2 Methods

Cyclic siloxanes and oxidized cyclic siloxanes were modeled with the 3D atmospheric chemical transport model CMAQ (Byun and Schere, 2006) modified to include cyclic siloxane species. CMAQ version 4.7.1 was used and the modeling domain covered the contiguous U.S., northern Mexico, and southern Canada. The domain had 14 vertical layers and a horizontal resolution of 36 km. Four, one-month simulations were performed for January, April, July, and October to characterize seasonal variability of cyclic siloxane atmospheric concentrations. A spin up period of 7 days was used to minimize the influence of zero initial conditions for the cyclic siloxanes species. Meteorology was from the Weather Research and Forecasting (WRF) model version 3.1.1 for the meteorological year of 2004.

The cyclic siloxanes were added to the CMAQ model by adding $D_4$, $D_5$, $D_6$, and the oxidized species, o-$D_4$, o-$D_5$, and o-$D_6$ to the cb05cl_ae5_aq mechanism. Rate constants for cyclic siloxanes reacting with hydroxyl radicals (OH) were used from Atkinson (1991) where $D_4$ and $D_5$ were determined experimentally and $D_6$ estimated from the reported $D_5$ per methyl rate. The rate constants used were $1.01 \times 10^{-12}$, $1.55 \times 10^{-12}$, and $1.92 \times 10^{-12}$ cm$^3$ molecule$^{-1}$ s$^{-1}$ for $D_4$, $D_5$, and $D_6$, respectively.

Wet and dry deposition of the primary species (e.g. $D_4$, $D_5$) were added to the model using Henry's law coefficients (Xu and Kropscott, 2012). For the oxidized cyclic siloxanes, physicochemical parameters were estimated using EPI Suite HENRYWIN v3.20 (EPA, 2012) for the single OH substitution of one methyl group of the parent cyclic siloxane (e.g., $D_3$TOH, $D_4$TOH). Deposition related inputs necessary for the CMAQ deposition routine included Henry's law coefficients, mass diffusivities, reactivity, and mesophyll resistance. The mass diffusivity calculations were calculated according to the Fuller, Schettler, and Giddings (FSG) method (Lyman et al., 1982) where molar volume was estimated based on element contributions. Sulfur molar volume contribution values were substituted for silicon atoms since silicon values were not





available. Calculated mass diffusivity values, as estimated by the FSG method were 0.0512 ($D_4$), 0.0454 ($D_5$), 0.0411 ($D_6$), 0.0527 (o-$D_4$), 0.0464 (o-$D_5$), and 0.0419 (o-$D_6$) $cm^2 s^{-1}$. The reactivity parameter was set at 2.0 in common with methanol and other species of limited reactivity. The mesophyll resistance, which is used to account for uptake by plants, was set to zero (only a few species had mesophyll resistances specified in CMAQ, such as $NO_2$, NO, CO, and Hg gas). Molecular weight

for the oxidized cyclic siloxanes assumed the single substituted OH species. The molecular weight of $D_6$ and o-$D_6$ exceeded the limit of the CMAQ dry deposition routine m3dry (390 g $mol^{-1}$) and values in excess of the limit were coerced to the limit. The impact of this substitution is expected to be minimal, since it is a minor adjustment to a minor pathway; dry deposition of cVMS is relatively small (McLachlan et al., 2010; Xu and Wania, 2013; Whelan et al., 2004).

       Emissions of cyclic siloxanes were distributed according to gridded population for the U.S., Canada, and Mexico
while Caribbean countries were neglected. Cyclic siloxane emission rates were calculated from an industry derived country specific $D_5$ emission rate for U.S., Canada, and Mexico (Buser et al., 2014; van Egmond, 2013). The method of estimating the $D_5$ emission rates has been described previously (McLachlan et al., 2010), but briefly, country specific antiperspirant sales data was combined with 2009 consumption data. To calculate $D_4$ and $D_6$ emission rates, ambient measurements from Chicago (Yucuis et al., 2013) were used to estimate emission ratios relative to $D_5$. Since OH reactivity (and other fate and transport

properties) vary from compound to compound, ambient measurements of compound ratios will not match emission ratios, except in air parcels that are so fresh as to have seen no oxidation. To check for the influence of air mass aging in the measurements of Yucuis et al. (2013), the ratio $NO_x/NO_y$ was used as a marker of air mass age (Slowik et al., 2011). This ratio is high in fresh emissions, and decreases as the air mass is oxidized. Hourly measurements of $NO_x$ and $NO_y$ from Northbrook, Illinois (EPA) were inspected during the time period of the Chicago sampling in Yucuis et al. (2013). Using the

$NO_x/NO_y$ photochemical age estimate, we calculated that emitted ratios vs. ambient ratios likely differed by less than 1% (see Supplemental section). The Chicago cyclic siloxane measurements were therefore used as emission ratios without photochemical age correction. The resulting emission ratios, 0.243 and 0.0451 for $D_4/D_5$ and $D_6/D_5$ respectively, were multiplied by the $D_5$ emission rate to estimate the $D_4$ and $D_6$ emission rates. The resulting $D_4$, $D_5$, and $D_6$ country emission rates were multiplied by gridded population and merged with Sparse Matrix Operator Kernel Emissions (SMOKE) model

version 2.5 generated year 2004 emissions. Cyclic siloxane emissions were constant for all simulations.

       Boundary conditions were from previous modeling (Danish Eulerian Hemispheric Model, DEHM) that modeled $D_5$ concentrations using 2009 $D_5$ emission rates as described above (Hansen et al., 2008; McLachlan et al., 2010). The DEHM model was run for the Northern Hemisphere at 150 km resolution. We extracted the $D_5$ concentrations from the DEHM model for year 2011 meteorology along our model boundary. Boundary concentrations were horizontally and vertically resolved,

varied by month, but were time invariant within each month. Since the DEHM model only included $D_5$, $D_4$ and $D_6$ concentrations were estimated using measurement ratios taken from a background site at Point Reyes, CA (Genualdi et al., 2011). Point Reyes samples had ratios of 0.646 and 0.0877 for $D_4/D_5$ and $D_6/D_5$ respectively. The background ratios combined with the "fresh" emission ratios (described previously) were used to calculate a photochemical age. The calculation of a photochemical age was necessary since the siloxanes have different OH reaction rates and therefore the siloxane ratios change



with season due to varying OH concentrations. Using this method we calculated an age of 17.6 days using the $D_4/D_5$ ratios and this is the age used for further calculations. The calculated photochemical age was then combined with season specific OH concentrations (Spivakovsky et al., 2000) to calculate monthly resolved $D_4/D_5$ and $D_6/D_5$ "background" ratios. These monthly resolved $D_4/D_5$ and $D_6/D_5$ ratios were then used for the entire model boundary. Additional details are available in the

Supplemental section.

## 3 Results and Discussion

### 3.1 Spatial Variation in Concentrations

Figures 1 and 2 show the 30 day averaged $D_5$ and oxidized $D_5$ (o-$D_5$) modeled concentrations for January, April, July, and October. The spatial distribution of cVMS and oxidized cVMS compounds show a strong population dependence with major

urban areas having elevated $D_5$ concentrations and peak o-$D_5$ concentrations occurring hundreds of km downwind of source regions due to the time it takes for the parent compounds to react with OH. Table 1 displays the monthly minimum, maximum, and average for the entire modeled domain. The 36-km grid cell with the highest 30-day average surface concentration of $D_5$ was 432, 379, 301, and 265 ng m$^{-3}$ for January (Los Angeles – Long Beach), April (Los Angeles – Long Beach), October (New York City), and July (New York City), respectively. The domain-averaged surface concentrations of $D_5$ were 6.82, 6.43, 5.09,

and 4.04 ng m$^{-3}$ for January, October, April, and July. Simulated o-$D_5$ was much lower than simulated $D_5$ concentrations. For example, the 36-km grid cell with the highest 30-day average surface concentration of o-$D_5$ was 9.04, 5.21, 4.86, and 3.19 ng m$^{-3}$ for July (NE of Los Angeles – Victorville), October (E of Los Angeles – San Bernardino), April (SE of Los Angeles – Mission Viejo), and January (Los Angeles – Long Beach), respectively. The domain average surface concentration for o-$D_5$ was 0.81, 0.72, 0.63, 0.37 ng m$^{-3}$ for July, April, October, and January, respectively. The peak domain-averaged concentrations

occurred during January for $D_5$ and July for o-$D_5$ which is expected based on seasonal trends of OH in North America (Spivakovsky et al., 2000).

Tables 2 and 3 show the monthly averaged cVMS and oxidized cVMS concentrations for 26 U.S. and Canadian sites. These sites include the most populous ten U.S. metropolitan areas, siloxane measurement sites, and NOAA Climate Monitoring and Diagnostics Laboratory (CMDL) sites. Modeled concentrations are strongly dependent on population with New York City

and Los Angeles having the highest concentrations. In addition to the population dependence, concentrations were greatest for $D_5$ followed by $D_4$ and $D_6$. This follows from our assumed emission ratios and agrees with North American measurement data (Yucuis et al., 2013; Genualdi et al., 2011; Ahrens et al., 2014; Krogseth et al., 2013b). The prevalence of $D_4$ relative to $D_6$ is of interest because analysis of cVMS composition in consumer products (Horii and Kannan, 2008; Wang et al., 2009; Dudzina et al., 2014; Lu et al., 2011; Capela et al., 2016) suggests that $D_6$ is more abundant than $D_4$ – while in our modeling

(and atmospheric measurements) $D_4$ concentrations are higher than $D_6$ concentrations. Four explanations bear further investigation: (1) $D_4$ may have non-negligible emissions from sources other than personal care products (e.g. industrial uses which are not captured in current emission estimates), (2) possible siloxane conversion during sample collection (Kierkegaard





and McLachlan, 2013; Krogseth et al., 2013a), (3) higher $D_4$ volatility (Lei et al., 2010) could cause both more difficult detection in personal care products and a larger fraction volatilization from products, and (4) uncertainty in the $D_4/D_6$ ratio taken from ambient measurements in Chicago to extend the $D_5$ emissions estimates to $D_4$ and $D_6$.

## 3.2 Seasonal Variation in Concentrations

Since OH concentrations vary seasonally we expect higher cVMS in the winter (low OH) and lower in the summer (high OH). This has been supported by previous measurement studies. For example, McLachlan et al. (2010) measured $D_5$ at a rural site in Sweden (59°N) and observed reduced $D_5$ concentrations for the period of May-June compared to January-April. Measurements in a remote Artic location (79°N) observed higher concentrations in the winter compared to the fall (Krogseth et al., 2013a). For OH concentrations to influence cVMS concentrations, time for oxidation is required – so the relationship

between seasonal OH and cVMS is expected at receptor sites where most cVMS is transported from upwind locations. At source-dominated locations, the influence of OH should be limited. For example, studies from Toronto highlight local meteorological influences as important in determining variation in siloxane ($D_3$-$D_6$) concentrations (Ahrens et al., 2014; Krogseth et al., 2013b).

    Figure 1 shows similar $D_5$ spatial distribution between the four months, especially for urban areas. Domain peak and

average concentrations (Table 1) have highest concentrations in January and lowest in July which agree with seasonal OH concentrations but specific grid cells (particularly urban locations) often deviate from this. Rural and remote locations are more likely to follow the OH-induced seasonal pattern. Seasonal variation for the 26 sites in Table 2 was examined using the most prevalent month highest concentrations occurred. Sites were classified as either urban or rural based on summer $D_5$ concentrations. For urban sites, the most prevalent month with highest average $D_5$ concentration was October (59%), followed

by July (23%), and January (18%). Restricting the analysis to the rural sites (summer $D_5$ concentration below 17 ng m$^{-3}$), peak $D_5$ concentrations occurred in January (56%), followed by October (33%), and April (11%). The month of lowest average $D_5$ concentrations occurred in July for 100% of the rural sites and 24% of the urban sites. Similarly, looking at the breakdown for the monthly averaged oxidized $D_5$ concentrations, highest concentrations generally occurred in July, which was true for 73% of the 26 sites. Figure 2 shows significant differences in the spatial distribution of o-$D_5$ between months. The analyzed

sites therefore suggest less of a seasonal trend for the parent compounds as compared to the oxidized products, and there are differences in seasonal trends between source and non-source locations. Remote and rural sites are more dependent on lifetime with respect to reaction with OH, while source locations are less sensitive. This agrees with previous modeling which showed reduced seasonal variability of $D_5$ concentrations for urban areas compared to remote locations (McLachlan et al., 2010; MacLeod et al., 2011; Xu and Wania, 2013).

Statistical relationships between $D_5$, OH, planetary boundary layer (PBL) height, and wind speed (WS) were explored using least squares multiple linear regression. For the 26 analyzed sites, OH, PBL, and WS values were normalized to their summer values and then used as predictive variables of the ratio of $D_5$ in each season to its summer value at the same site. Sites were split between urban and rural as described previously. For urban sites, $D_5$ concentration was only correlated to OH$^-$



[1] when WS$^{-1}$ was also included, with WS being the dominant variable. The strongest predictive variables were PBL$^{-1}$ and WS$^{-1}$ with an adjusted R$^2$ fit of 0.50 and a p-value of <0.001. The regression analysis supports the previous conclusion: ventilation of local emissions through PBL height and local winds is the strongest influence on urban siloxane concentrations.

For the rural sites, WS$^{-1}$ was the only variable of significance but had a low adjusted R$^2$ of 0.10, p-value of 0.056, and a negative coefficient meaning lower wind speed results in lower D$_5$ concentrations. Repeating the linear regression excluding Canadian sites and Point Reyes (CA), led to similar results. Canadian sites were excluded since non-siloxane Canadian emissions were allocated by population and may cause errors in OH due to misallocation of nitrogen oxides and reactive organic gases from some source sectors (Spak et al., 2012). Point Reyes was excluded due to high grid cell population despite low D$_5$ concentrations. See the Supplemental section for additional regression results. From this analysis, we conclude that factors other than local OH and local meteorology control rural/remote siloxane concentrations. These factors likely include regional OH and regional transport patterns.

### 3.3 Model-Measurement Comparison

The model results were compared to measurement values in the Midwest (Yucuis et al., 2013), North American measurements from the Global Atmospheric Passive Sampling (GAPS) network (Genualdi et al., 2011), and several Toronto measurements (Genualdi et al., 2011; Ahrens et al., 2014; Krogseth et al., 2013b).

### 3.3.1 Midwest Model Comparison

In Yucuis et al. (2013) measurements were taken at three Midwest locations during the summer (June-August) of 2011. The measurements were compared to the July modeled hourly concentrations that were averaged so the periods were of similar duration to the measurement sampling periods. The modeled period does not correspond to the exact measurement days or meteorology, but should be representative of typical summer concentrations. Measurements are from 2011 and the model's meteorological fields are from 2004; however, average wind speeds, wind directions, and boundary layer heights are typically similar from year to year.

Figure 3 displays the boxplot comparison of the three Midwest sites of Yucuis et al. (2013) and the modeled concentrations. The model does capture the population dependence that the measurements show, with Chicago observing highest concentrations followed by Cedar Rapids and West Branch. Modeled concentrations however are lower for all three locations compared to the measurements with fractional bias (Table S8) at Chicago of -0.31, -0.31, -0.28 (for D$_4$, D$_5$, D$_6$ respectively), Cedar Rapids -1.25, -0.93, -1.51, and West Branch -1.25, -0.78, -1.23. Comparing the relative percent error of the mean modeled concentrations to the measured values, Chicago sites had relative percent errors of around 25% while the other sites had values ranging from 56% - 86%. For Chicago, error between the species was similar and this is most likely the result that D$_4$ and D$_6$ emission rates were calculated based on the Chicago measurements. For Cedar Rapids and West Branch, D$_5$ had the lowest error while D$_4$ and D$_6$ were larger. This may indicate that the siloxane emission ratios vary based on location.





One possible explanation for low model concentrations could be low emission estimates. Current emission estimates (Table S2) vary considerably and the estimates used in this work were 32.8, 135, and 6.10 mg person$^{-1}$ day$^{-1}$ for $D_4$, $D_5$, and $D_6$ respectively for the U.S. and Canada, while the Mexico emissions were 5.92, 24.4, and 1.10 mg person$^{-1}$ day$^{-1}$ for $D_4$, $D_5$, and $D_6$. Previous emission estimates have ranged from $0.001 - 100$, $0.002 - 1200$, and $0.0009 - 80$ mg person$^{-1}$ day$^{-1}$ for $D_4$,

$D_5$, and $D_6$ respectively (Tang et al., 2015; Buser et al., 2013a; Buser et al., 2014; Navea et al., 2011; Yucuis et al., 2013; Horii and Kannan, 2008; Dudzina et al., 2014; Wang et al., 2009; Capela et al., 2016). Additionally, there could be other sources of siloxane emissions besides personal care products, or seasonal/regional differences that are not captured in current emission estimates.

### 3.3.2 GAPS Model Comparison

The model was also compared to measurements of Genualdi et al. (2011). These measurements were collected from passive samplers as part of the GAPS network over three months in 2009, generally from late March to early July. Figure 4 shows the CMAQ modeled April month versus measurements for eight locations within our domain. Again as with the Yucuis et al. (2013) comparison, the modeled results do not explicitly represent meteorological conditions of the measurement period. Fractional error (Table S9) for $D_4$ varied from $0.02 - 1.93$ with Point Reyes having the lowest and Ucluelet the highest. For

$D_5$, fractional error values ranged from $0.02 - 1.24$ with Fraserdale the lowest and Bratt's Lake the highest. Similarly, for $D_6$, the fractional error varied from $0.11 - 1.71$ with Bratt's Lake the lowest and Ucluelet the highest. Averaged over the eight sites, the overall fractional biases were -0.41, -0.03, and -0.90 for $D_4$, $D_5$, and $D_6$, respectively. The mean fractional error was 0.95, 0.66, and 0.98 for $D_4$, $D_5$, and $D_6$ species. Therefore, based on the fractional error values, $D_5$ had the best agreement followed by $D_4$ and $D_6$. This is not surprising that $D_5$ had the best agreement since $D_4$ and $D_6$ emission rates are estimated

based on Chicago measurements and would have additional uncertainty compared to the $D_5$ emission uncertainty.

On average, fractional bias for $D_5$ was close to zero while $D_4$ and $D_6$ had greater negative bias due to significant deviations for Fraserdale, Ucluelet, and Whistler. Aside from these three sites, the $D_4$ predictions generally agreed well with the measurements. These same three sites and Groton were also significantly under predicted for $D_6$ but other sites were within a factor of 2 of the measurements. Possible explanations for model deviation could be population errors (Ucluelet and Whistler

experience seasonal tourism), product transformation of higher molecular weight siloxanes to $D_4$ on sampling media (Kierkegaard and McLachlan, 2013; Krogseth et al., 2013a), or our boundary conditions could be underestimating Asian cVMS transport. Genualdi et al. (2011) hypothesized the high $D_4$ concentrations measured at Whistler and Ucluelet could be due to transport from Asia since $D_4$ concentrations were greatest at west coast locations and especially at high altitude sites.

Model overprediction for $D_5$ occurred for the Point Reyes and Bratt's Lake sites. Representation error is a likely cause

of this, since the actual sampling sites were upwind of large population centers (San Francisco and Regina, SK) in these grid cells; at 36 km resolution, the upwind sampling sites and the downwind emission centers are not resolved. However, Point Reyes and Bratt's Lake $D_4$ and $D_6$ concentrations were close to the modeled values.



We also compare the 36 km CMAQ $D_5$ concentration results to values from the DEHM and BETR models. The BETR model did not report values for Ucluelet or Groton so those sites are not included. The $D_5$ modeling attempts were ordered from most skilled to least skilled by using the mean of the fractional bias and fractional error (in parenthesis) scores, CMAQ -0.03 (0.66), DEHM -0.53 (0.73), and BETR -0.81 (1.08). The CMAQ and DEHM models had similar performance

for Fraserdale, Whistler, Ucluelet, and Point Reyes, while the urban areas (Downsview, Sydney (FL), and Groton) were better predicted in the CMAQ model. Bratt's Lake was overestimated compared to the DEHM model and may have to do with the greater influence of Regina, SK emissions due to improved model resolution. The differences in modeled concentrations are most likely due to higher spatial resolution for CMAQ (36 km) versus 150 km (DEHM), and 15° (BETR) resolutions.

### 3.3.3 Toronto Model Comparison

Multiple measurement and modeling studies have investigated cVMS concentrations in Toronto, Canada. Table 4 shows the mean and hourly range of cVMS concentrations in Toronto for each of the four months as simulated by the CMAQ model. Table 4 further includes the March 2010 – April 2011 measured concentrations as collected by both passive and active sampling (Ahrens et al., 2014), active sampling from March 2012 – June 2012 (Krogseth et al., 2013b), and passive sampling (April – June 2009) from the GAPS network (Genualdi et al., 2011). Finally, the BETR and DEHM modeled $D_5$ concentrations (Apr

– Jun 2009) are also tabulated (Genualdi et al., 2011). The CMAQ results compared favorably to the Ahrens et al. (2014) measurements with CMAQ monthly averages that generally fell within the reported measurement concentration ranges. $D_4$ monthly averages were within a factor of 0.97 – 1.94, $D_5$ within a factor of 0.59 – 1.39, and $D_6$ within a factor of 0.33 – 0.78 of the yearly averaged active and passive sampling measurements. Comparing the range of concentrations, CMAQ hourly ranges were 1.8 – 110.3 ($D_4$), 6.0 – 453.1 ($D_5$), and 0.24 – 20.42 ($D_6$) ng m$^{-3}$ compared to Ahrens et al. (2014) 24-hour active

sampling range of 2.8 – 77 ($D_4$), 15 – 247 ($D_5$), and 1.9 – 22 ($D_6$) ng m$^{-3}$. The greater modeled range can likely be attributed to hourly concentrations as opposed to 24-hour. Similarly, good agreement was observed for the measurements from Krogseth et al. (2013b), average April CMAQ $D_4$, $D_5$, and $D_6$ concentrations were a factor of 0.84, 0.88, and 0.67 respectively of the measured average. The range of concentrations were similar compared to the April CMAQ month, with the measurements having higher peak concentrations despite a longer sampling time (2-3 days). While CMAQ April averages were 1.85, 1.49,

and 0.59 times the Genualdi et al. (2011) measurements. Previous Toronto modeling predicted 6.5 ng m$^{-3}$ (BETR) and 28 ng m$^{-3}$ (DEHM) which were significantly lower than the spring CMAQ $D_5$ concentration of 81.6 ng m$^{-3}$. Overall, the CMAQ model was able to better predict the higher observed concentrations of Toronto, which again, can most likely be attributed to increased model resolution.

### 3.4 Compound Ratios

Cyclic siloxane product ratios can be used to gain insight into emission sources and OH photochemical aging (Ahrens et al., 2014; Kierkegaard and McLachlan, 2013; Krogseth et al., 2013b; Krogseth et al., 2013a; Yucuis et al., 2013; Navea et al., 2011). Figures 5 and 6 show the seasonal plots of monthly averaged $D_5/D_4$ and $D_6/D_5$ product ratios. Due to differences in



OH reactivity rates, cyclic siloxane reactivity increases with Si-O chain length (more methyl groups) so that $D_6$ is the most reactive and $D_4$ the least (Atkinson, 1991). Therefore, siloxane ratios depend on emissions, exposure to OH, and relative reactivity rates. Figures 5 and 6 display the mole ratios with the more reactive species as the numerator; as air masses move away from emission sources and are exposed to OH, the ratio decreases due to more rapid depletion of the more reactive species. This is evident in the $D_5/D_4$ and $D_6/D_5$ maps which show urban areas have the highest ratios.

Seasonal differences of the product ratios are similar for both $D_5/D_4$ and $D_6/D_5$ mole ratios. Urban areas exhibit almost no season-to-season difference (Table S5), as they reflect the local emission ratios. Seasonal differences are most apparent for rural and remote locations. Domain average ratios are highest in January and lowest in July which is consistent with seasonal OH fluctuations.

Since both $SO_2$ and cVMS are precursors to secondary aerosol formation, and both compounds have approximately the same OH rate constant, the ratio of gas phase $SO_2$ to cVMS should predict aerosol-phase ratios of S to Si in photochemically generated particles (Bzdek et al., 2014). Figure 7 shows the seasonally modeled, monthly averaged gas phase $SO_2/(D_4 + D_5 + D_6)$ mole ratios. Urban ratios exhibit lowest values which suggest photochemically generated aerosols would have increased Si composition derived from siloxane oxidation. Conversely, rural locations have high $SO_2$/cVMS ratios and expected low Si aerosol composition. This is consistent with the high nanoparticle Si measured in Pasadena, CA and Lewes, DE by Bzdek et al. (2014). Seasonal variation in the $SO_2$/cVMS ratio is minor.

### 3.5 Vertical profile analysis

Modeled monthly averaged $D_5$ and o-$D_5$ vertical profiles are shown in Figure 8 for three grid cells near Los Angeles. The locations of the analyzed sites include the highest monthly averaged surface July $D_5$ concentration, the highest averaged surface o-$D_5$ concentration, and a grid cell over the Pacific Ocean. The grid cell with greatest $D_5$ concentration, (termed "Peak $D_5$") included cities such as Long Beach and Anaheim while the grid cell with highest o-$D_5$ ("Peak o-$D_5$") was approximately 80 km northeast of the peak $D_5$ grid cell and included Victorville and Hesperia, CA. The third location was over the Pacific Ocean ("Pacific") approximately 195 km southwest of Los Angeles (Fig. S9).

The CMAQ model was run with 14 vertical layers; plotted is the layer top height versus the monthly averaged July $D_5$ and o-$D_5$ concentration. For $D_5$ concentrations, both the "Peak $D_5$" and "Peak o-$D_5$" sites had highest concentrations at the surface. Over the Pacific, concentrations peaked above the surface at approximately 700-1,700 m. Surface $D_5$ concentrations were 251, 103, and 0.3 ng m$^{-3}$ for the "Peak $D_5$", "Peak o-$D_5$", and "Pacific" locations respectively. From heights 475-3,000 m, the "Peak o-$D_5$" site had higher $D_5$ concentrations than the "Peak $D_5$" site and this is most likely due to the plume dilution from the upwind LA source. For o-$D_5$ concentrations, surface concentrations were highest for the "Peak o-$D_5$" site (9 ng m$^{-3}$), followed by the "Peak $D_5$" site (2 ng m$^{-3}$), and the "Pacific" site (0.2 ng m$^{-3}$). From the surface to 3,000 m the "Peak o-$D_5$" grid cell had highest o-$D_5$ concentrations as a result of being downwind of a major emission source and the oxidation reaction takes times to occur. Both the "Peak $D_5$" and "Pacific" sites have peak o-$D_5$ concentrations not at the surface (475 and 2,300 meters respectively) while the "o-$D_5$" site is at the surface. The low surface o-$D_5$ at the peak $D_5$ site could be due to low OH




concentrations caused by urban OH sinks and is consistent with low modeled surface OH (Fig. S10). Vertical concentrations appear to be dependent on transport, reaction time, and OH concentrations.

## 4 Conclusions

The CMAQ model was modified to include $D_4$, $D_5$, $D_6$, and the oxidation products to investigate urban-rural concentration
5   gradients, seasonal variability, product and $SO_2$ mole ratios, and vertical profiles. Improved model performance was observed when compared to previous modeling especially for urban areas. Concentrations are heavily dependent on population with strong urban/rural concentration gradients observed. Urban areas have highest cVMS concentrations but are not significantly influenced by seasonal variability of OH, while rural cVMS is influenced by transport and regional OH concentrations. The oxidized product concentrations are significantly lower than the parent compounds with average $D_5$ concentrations up to 432
10   ng m$^{-3}$ and average o-$D_5$ up to 9 ng m$^{-3}$. Highest oxidized siloxane concentrations occur downwind of major urban centers. Increased error for modeled $D_4$ and $D_6$ relative to $D_5$ is due to uncertainty in emission estimates. Future work should address these emission uncertainties by exploring seasonal, temporal, spatial, and non-personal care product emissions.

While the parent compounds have been extensively studied, the environmental and health impact of the oxidized species have not been addressed. This is especially important since the oxidation products likely form particles. To the best
of our knowledge this work provides the first estimated atmospheric loadings and spatial distribution of the oxidized species. Future work should focus on gas and particle phase measurements of the oxidized species to confirm particle formation in the ambient environment and to determine typical loadings in the environment. This is especially important since exposure would be expected to be highest indoors where cyclic siloxane concentrations are greatest.

### Acknowledgements

This research was funded by the National Institute of Environmental Health Sciences through the University of Iowa Environmental Health Sciences Research Center, NIEHS/NIH P30ES005605; and by Iowa Superfund Research Program, National Institute of Environmental Health Sciences Grant P42ES013661. We want to thank Jaemeen Baek (formerly University of Iowa) for providing the model meteorology and non-siloxane emissions. We also want to thank Scott Spak (University of Iowa) for his CMAQ guidance.

**Competing Interests**

The authors declare that they have no conflict of interest.



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





**Figure 1: Monthly averaged surface layer D$_5$ concentrations. The domain average concentration is shown in the lower left for each month.**



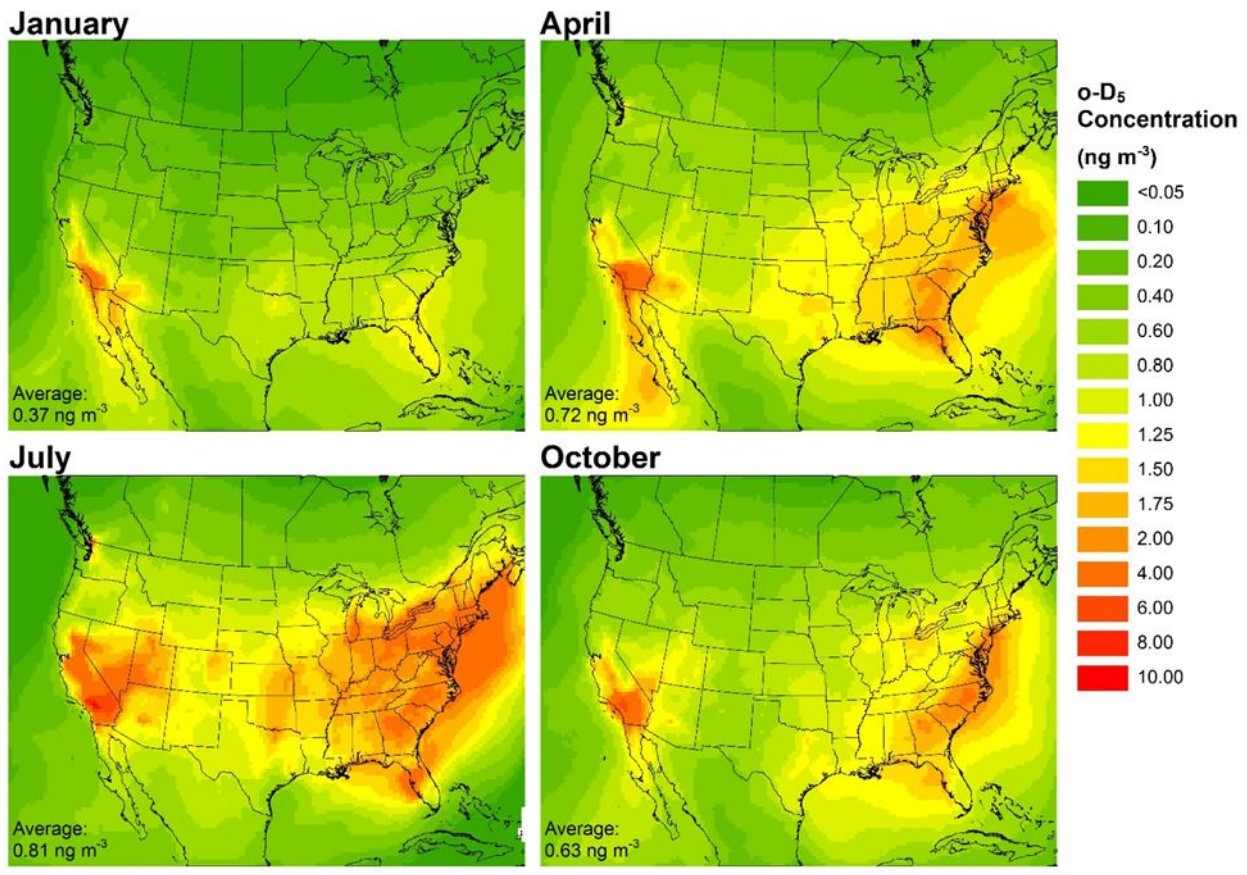

**Figure 2:** **Monthly average surface layer oxidized D₅ (o-D₅) concentrations. The domain average concentration is shown in the lower left for each month.**





**Table 1:** **Monthly minimum, maximum, and average $D_5$ and $o\text{-}D_5$ concentrations in the lowest modeled layer for the domain.**

| Domain | $D_5$ Concentrations (ng m$^{-3}$) | | | | $o\text{-}D_5$ Concentrations (ng m$^{-3}$) | | | |
|---|---|---|---|---|---|---|---|---|
| | January | April | July | October | January | April | July | October |
| Minimum | 0.14 | 0.27 | 0.024 | 0.27 | 0.0031 | 0.037 | 0.0021 | 0.0033 |
| Maximum | 432 | 379 | 265 | 301 | 3.19 | 4.86 | 9.04 | 5.21 |
| Average | 6.82 | 5.09 | 4.04 | 6.43 | 0.37 | 0.72 | 0.81 | 0.63 |





**Table 2:** Average monthly CMAQ modeled surface cVMS concentrations (ng m⁻³) sorted by population (highest at top of table) in analyzed grid cell. Minimum and maximum values in each column in boldface and italicized.

| Site | D₄ January | April | July | October | D₅ January | April | July | October | D₆ January | April | July | October |
|---|---|---|---|---|---|---|---|---|---|---|---|---|
| New York, NY, USA | 57.3 | 55.9 | *64.9* | *73.6* | 234 | 228 | *265* | *301* | 10.5 | 10.2 | *11.9* | *13.5* |
| Los Angeles, CA, USA | *105* | *92.6* | 61.3 | 65.1 | *432* | *379* | 251 | 266 | *19.4* | *17.0* | 11.3 | 12.0 |
| Chicago, IL, USA | 34.1 | 32.4 | 40.9 | 40.1 | 139 | 132 | 168 | 164 | 6.26 | 5.92 | 7.56 | 7.37 |
| Pasadena, CA, USA | 38.9 | 39.0 | 48.6 | 39.5 | 159 | 159 | 198 | 161 | 7.15 | 7.11 | 8.90 | 7.21 |
| Houston, TX, USA | 28.4 | 25.8 | 30.0 | 25.9 | 116 | 105 | 123 | 106 | 5.23 | 4.73 | 5.55 | 4.76 |
| Washington, DC, USA | 29.4 | 30.0 | 35.1 | 43.5 | 120 | 122 | 144 | 178 | 5.38 | 5.46 | 6.47 | 8.01 |
| Miami, FL, USA | 28.1 | 17.0 | 20.9 | 24.3 | 115 | 69.1 | 85.7 | 99.3 | 5.17 | 3.10 | 3.86 | 4.47 |
| Boston, MA, USA | 21.0 | 21.3 | 25.9 | 26.1 | 84.9 | 85.4 | 105 | 106 | 3.79 | 3.81 | 4.69 | 4.74 |
| Downsview, ON, CAN | 21.7 | 20.2 | 28.2 | 30.9 | 88.0 | 81.6 | 115 | 126 | 3.94 | 3.65 | 5.19 | 5.64 |
| Atlanta, GA, USA | 24.8 | 21.2 | 23.4 | 27.1 | 101 | 86.0 | 95.6 | 111 | 4.54 | 3.86 | 4.30 | 4.98 |
| Philadelphia, PA, USA | 21.7 | 21.7 | 21.3 | 30.4 | 88.2 | 87.3 | 86.3 | 124 | 3.95 | 3.90 | 3.86 | 5.54 |
| Dallas, TX, USA | 20.5 | 15.6 | 12.8 | 22.9 | 83.5 | 63.4 | 52.1 | 93.5 | 3.75 | 2.84 | 2.34 | 4.20 |
| Sydney, FL, USA | 12.5 | 10.2 | 12.6 | 11.0 | 50.8 | 40.7 | 50.6 | 44.7 | 2.27 | 1.81 | 2.25 | 2.00 |
| Cedar Rapids, IA, USA | 4.91 | 4.06 | 4.37 | 5.88 | 19.4 | 15.5 | 17.4 | 23.3 | 0.853 | 0.675 | 0.777 | 1.03 |
| Point Reyes, CA, USA | 8.04 | 4.12 | 2.10 | 4.63 | 32.6 | 16.1 | 8.38 | 18.6 | 1.46 | 0.707 | 0.373 | 0.826 |
| Bratt's Lake, SK, CAN | 2.86 | 2.25 | 1.88 | 2.45 | 11.2 | 8.15 | 7.24 | 9.53 | 0.492 | 0.348 | 0.320 | 0.416 |
| Groton, CT, USA | 7.62 | 11.2 | 11.3 | 7.91 | 30.0 | 43.9 | 44.3 | 30.8 | 1.32 | 1.93 | 1.95 | 1.34 |
| Lewes, DE, USA | 6.99 | 6.67 | 5.31 | 8.61 | 27.6 | 25.6 | 20.8 | 34.0 | 1.22 | 1.12 | 0.915 | 1.50 |
| Harvard Forest, MA, USA | 6.06 | 5.55 | 5.93 | 7.06 | 23.5 | 20.8 | 22.9 | 27.5 | 1.03 | 0.901 | 1.01 | 1.20 |
| West Branch, IA, USA | 3.42 | 2.46 | 2.28 | 4.66 | 13.2 | 8.88 | 8.82 | 18.3 | 0.576 | 0.378 | 0.389 | 0.804 |
| Whistler, BC, CAN | 1.39 | 1.30 | 0.728 | 1.11 | 5.40 | 4.47 | 2.73 | 4.21 | 0.235 | 0.185 | 0.118 | 0.181 |
| Trinidad Head, CA, USA | 1.55 | 0.900 | 0.626 | 0.852 | 6.03 | 2.88 | 2.35 | 3.11 | 0.263 | 0.115 | 0.102 | 0.131 |
| Park Falls, WI, USA | 1.67 | 1.19 | 0.911 | 2.23 | 5.91 | 3.59 | 3.12 | 8.31 | 0.242 | 0.138 | 0.131 | 0.354 |
| Niwot Ridge, CO, USA | *0.478* | 0.829 | 0.468 | *0.649* | *1.77* | 2.82 | 1.54 | *2.35* | *0.0749* | 0.116 | 0.0623 | *0.0985* |
| Ucluelet, BC, CAN | 1.66 | *0.827* | *0.142* | 0.687 | 6.46 | 2.46 | *0.423* | 2.42 | 0.282 | 0.0932 | *0.0170* | 0.0992 |
| Fraserdale, ON, CAN | 1.06 | 0.869 | 0.350 | 1.71 | 2.88 | *1.93* | 0.756 | 5.88 | 0.0929 | *0.0559* | 0.0250 | 0.237 |





**Table 3:** Average monthly CMAQ modeled surface oxidized cVMS concentrations (ng m$^{-3}$) sorted by population (highest at top of table) in analyzed grid cell. Minimum and maximum values in each column in boldface and italicized.

| | o-D$_4$ | | | | o-D$_5$ | | | | o-D$_6$ | | | |
|---|---|---|---|---|---|---|---|---|---|---|---|---|
| Site | January | April | July | October | January | April | July | October | January | April | July | October |
| New York, NY, USA | 0.0760 | 0.383 | *0.782* | 0.404 | 0.454 | 2.11 | *4.60* | 2.46 | 0.0249 | 0.112 | 0.254 | 0.135 |
| Los Angeles, CA, USA | *0.460* | *0.656* | 0.315 | 0.576 | *3.19* | *4.26* | 2.01 | 3.89 | *0.190* | *0.246* | 0.114 | 0.228 |
| Chicago, IL, USA | 0.0622 | 0.278 | 0.427 | 0.219 | 0.359 | 1.32 | 2.30 | 1.27 | 0.0191 | 0.0642 | 0.122 | 0.0681 |
| Pasadena, CA, USA | 0.389 | 0.655 | 0.720 | *0.666* | 2.49 | 3.92 | 4.46 | *4.21* | 0.142 | 0.215 | 0.248 | *0.237* |
| Houston, TX, USA | 0.133 | 0.274 | 0.244 | 0.212 | 0.776 | 1.42 | 1.35 | 1.24 | 0.0417 | 0.0731 | 0.0726 | 0.0673 |
| Washington, DC, USA | 0.0807 | 0.363 | 0.560 | 0.330 | 0.470 | 1.94 | 3.20 | 2.01 | 0.0253 | 0.101 | 0.175 | 0.112 |
| Miami, FL, USA | 0.206 | 0.233 | 0.151 | 0.166 | 1.28 | 1.20 | 0.824 | 1.01 | 0.0711 | 0.0612 | 0.0439 | 0.0564 |
| Boston, MA, USA | 0.0579 | 0.246 | 0.555 | 0.239 | 0.334 | 1.28 | 3.14 | 1.43 | 0.0180 | 0.0661 | 0.173 | 0.0779 |
| Downsview, ON, CAN | 0.0424 | 0.210 | 0.373 | 0.162 | 0.246 | 0.994 | 2.08 | 0.948 | 0.0132 | 0.0483 | 0.113 | 0.0514 |
| Atlanta, GA, USA | 0.176 | 0.402 | 0.464 | 0.344 | 1.06 | 1.99 | 2.52 | 2.03 | 0.0582 | 0.0989 | 0.134 | 0.110 |
| Philadelphia, PA, USA | 0.0724 | 0.389 | 0.607 | 0.353 | 0.429 | 2.12 | 3.48 | 2.14 | 0.0234 | 0.112 | 0.191 | 0.118 |
| Dallas, TX, USA | 0.179 | 0.286 | 0.299 | 0.221 | 1.04 | 1.39 | 1.54 | 1.26 | 0.0554 | 0.0687 | 0.0797 | 0.0666 |
| Sydney, FL, USA | 0.186 | 0.340 | 0.528 | 0.234 | 1.14 | 1.78 | 3.01 | 1.40 | 0.0630 | 0.0914 | 0.163 | 0.0767 |
| Cedar Rapids, IA, USA | 0.0531 | 0.233 | 0.264 | 0.154 | 0.295 | 1.00 | 1.29 | 0.864 | 0.0153 | 0.0454 | 0.0659 | 0.0455 |
| Point Reyes, CA, USA | 0.0639 | 0.113 | 0.0639 | 0.0898 | 0.434 | 0.627 | 0.405 | 0.588 | 0.0256 | 0.0336 | 0.0237 | 0.0343 |
| Bratt's Lake, SK, CAN | *0.00994* | 0.104 | 0.0971 | 0.0550 | *0.0547* | 0.362 | 0.394 | 0.282 | *0.00288* | 0.0134 | 0.0185 | 0.0138 |
| Groton, CT, USA | 0.0691 | 0.254 | 0.644 | 0.247 | 0.427 | 1.56 | 4.27 | 1.60 | 0.0240 | 0.0900 | *0.257* | 0.0922 |
| Lewes, DE, USA | 0.0759 | 0.340 | 0.478 | 0.291 | 0.449 | 1.85 | 2.69 | 1.78 | 0.0245 | 0.0989 | 0.147 | 0.0986 |
| Harvard Forest, MA, USA | 0.0506 | 0.224 | 0.394 | 0.199 | 0.292 | 1.12 | 2.15 | 1.16 | 0.0159 | 0.0562 | 0.117 | 0.0626 |
| West Branch, IA, USA | 0.0535 | 0.238 | 0.269 | 0.165 | 0.298 | 1.03 | 1.31 | 0.933 | 0.0155 | 0.0465 | 0.0669 | 0.0494 |
| Whistler, BC, CAN | 0.0146 | 0.0863 | 0.0745 | 0.0306 | 0.0838 | 0.375 | 0.386 | *0.170* | 0.00453 | 0.0170 | 0.0201 | *0.00889* |
| Trinidad Head, CA, USA | 0.0246 | 0.0742 | 0.0559 | 0.0516 | 0.164 | 0.345 | 0.340 | 0.329 | 0.00966 | 0.0163 | 0.0195 | 0.0189 |
| Park Falls, WI, USA | 0.0213 | 0.125 | 0.172 | 0.104 | 0.114 | 0.486 | 0.778 | 0.582 | 0.00578 | 0.0200 | 0.0387 | 0.0306 |
| Niwot Ridge, CO, USA | 0.0288 | 0.128 | 0.219 | 0.0994 | 0.156 | 0.569 | 1.00 | 0.549 | 0.00796 | 0.0265 | 0.0494 | 0.0285 |
| Ucluelet, BC, CAN | 0.0138 | 0.0662 | *0.0142* | *0.0273* | 0.0955 | 0.288 | *0.0499* | 0.177 | 0.00587 | 0.0125 | *0.00210* | 0.0102 |
| Fraserdale, ON, CAN | 0.0172 | *0.0557* | 0.0732 | 0.0668 | 0.0766 | *0.195* | 0.266 | 0.367 | 0.00328 | *0.00700* | 0.0119 | 0.0193 |



**Figure 3: Model comparison to Yucuis et al. (2013).** Model results are from CMAQ modeled July month. Hourly model data was averaged to 12, 24, and 36 hour periods to match sampling times as explained in the text. Median concentrations and number of observations are tabulated under the boxplots.





**Figure 4:** Comparison of the April averaged CMAQ model to Genualdi et al. (2011). BETR and DEHM model results are from Genualdi et al. (2011) and represent the same period as the measurements.





**Table 4: Toronto cyclic siloxane comparison between the CMAQ model and previous published measurement and modeling studies. Reported is the mean concentration with the range in parenthesis.**

| Period | Method | $D_4$ (ng m$^{-3}$) | $D_5$ (ng m$^{-3}$) | $D_6$ (ng m$^{-3}$) | Reference |
|---|---|---|---|---|---|
| | | Atmospheric Concentration, mean (range) | | | |
| January | CMAQ Model | 21.7 (2.0 - 77.1) | 88.1 (7.5 - 315.4) | 3.94 (0.31 - 14.19) | *This study* |
| April | CMAQ Model | 20.4 (1.9 - 79.1) | 82.1 (6.0 - 323.6) | 3.67 (0.24 - 14.56) | *This study* |
| July | CMAQ Model | 28.3 (2.3 - 110.3) | 115.9 (8.9 - 453.1) | 5.22 (0.39 - 20.42) | *This study* |
| October | CMAQ Model | 31.0 (1.8 - 102.7) | 126.3 (6.6 - 420.3) | 5.67 (0.28 - 18.90) | *This study* |
| March 2010 - April 2011 | Active sampling | 16 (2.8 - 77) | 91 (15 - 247) | 7.3 (1.9 - 22) | *Ahrens et al. (2014)* |
| March 2010 - April 2011 | Passive sampling | 21 (9.3 - 35) | 140 (89 - 168) | 11 (8.0 - 20) | *Ahrens et al. (2014)* |
| March 2012 - June 2012 | Active sampling | 24.2 (4.7 – 90.9) | 93.5 (22.4 – 355) | 5.5 (1.6 – 17.4) | *Krogseth et al. (2013b)* |
| April 2009 - June 2009 | Passive sampling | 11 | 55 | 6.2 | *Genualdi et al. (2011)* |
| April 2009 - June 2009 | BETR Model | - | 6.5 | - | *Genualdi et al. (2011)* |
| April 2009 - June 2009 | DEHM Model | - | 28 | - | *Genualdi et al. (2011)* |



**Figure 5: Modeled monthly averaged $D_5/D_4$ mole ratios by season. Larger cVMS species react faster with OH. More reactive species are in the numerator; therefore, ratios decrease with air mass age.**





**Figure 6: Modeled monthly averaged D6/D5 mole ratios by season. Larger cVMS species react faster with OH. More reactive species are in the numerator; therefore, ratios decrease with air mass age.**





**Figure 7: Modeled monthly averaged $SO_2/ (D_4+D_5+D_6)$ mole ratio by season.**





**Figure 8:** **Monthly averaged vertical profiles for grid cells near Los Angeles. Grid cells refer to the location of maximum July D$_5$, maximum July o-D$_5$, and a grid cell over the Pacific Ocean.**