# Peer review of "Comprehensive Atmospheric Modeling of Reactive Cyclic Siloxanes and Their Oxidation Products"

_Atmospheric Chemistry and Physics, 2016_

## Referee Comment (RC1) · Anonymous Referee #1 · 5 Feb 2017

General Comments

This is a strong paper. The methodology is of high quality, the discussion of the results is insightful, and the paper is very well written. Large scale modeling of the atmospheric fate of cVMS has been previously conducted by others using similar methods and the general principles governing atmospheric fate of these chemicals have been discussed. The major technical innovations in this paper are:

- A higher spatial resolution, which allows a better description of concentrations in large urban areas.

- The inclusion of oxidation products in the model, which allows the concentration fields

of the products to be explored.

The major contributions of this paper to understanding are:

- A more thorough discussion of how cVMS emission patterns and properties affect the horizontal, vertical and temporal variability in concentrations of cVMS and their primary oxidation products. The horizontal and temporal variability for cVMS has been discussed earlier, but not in this depth. The vertical variability for cVMS is novel, as is the discussion of the cVMS oxidation products.

- A comparison of the model results is made with a larger and more diverse set of measurements than has been presented previously attempted, lending more confidence to the modeling, in particular for D4 and D6.

A disappointment with this paper is that more effort was not devoted to exploring the atmospheric fate of the cVMS oxidation products. Although these model data represent one of the major technical innovations of the work, the vast majority of the discussion is devoted to discussing the native cVMS.

Specific Comments

Page 2, line 4: Here the authors take sides in an ongoing infected scientific/political debate. They support this position by citing papers representing just one side of the debate. This is unnecessary and does not contribute to the credibility of their work.

Page 2, lines 16-17: Here several references are given for work that was published in just one paper, even though that one paper is expressly stated at the beginning of the sentence.

Page 2, lines 24-27: I am surprised by the selection of information presented here, given that the second author has published simulations that give much more detailed insight into the latitudinal gradients, urban-remote gradients, seasonal patterns and sensitivity to processes and parameterizations on a hemispheric scale.

[Figure]

Page 3, line 4: McLachlan et al. (2010) had no measured data from summer and autumn. Incorrect citation.

Page 4, line 26 and paragraph that follows: It is not clear how dry deposition was modeled. Were the surfaces considered to be infinite sinks? Is this a reasonable assumption for the cVMS? I am not aware of evidence showing that cVMS are rapidly degraded on surfaces, except on soil when it is very dry. cVMS have log octanol-air partition coefficients of about 4-6, which suggests that surface media have a limited ability to soak up cVMS and that a partitioning equilibrium between the atmosphere and surfaces would be reached quickly. In this case, assuming that surfaces are infinite sinks would be a poor assumption. While I doubt that this will modify the conclusions of this study, it is important that the assumptions made in modeling deposition are clearly stated and justified.

Page 6, line 31 and elsewhere: The literature cited by the authors discusses other sources of cVMS emissions besides personal care products. For instance, Brooke et al. (2009c) list emissions of residuals in PDMS as being twice as important as personal care products for emissions of D4 to air.

Page 8, lines 20-22: This assumption depends on the number of measurements that were conducted and the time period over which the measurements integrated. A handful of very short measurements can well give a non-representative picture of "typical" summer concentrations. I suggest that the authors provide some information on the number and length of the measurements.

p. 10, lines 18-20: Why are hourly modeled data compared with daily measurements when it is clear that the different time scales makes the comparison difficult? I suggest that you integrate the modeled data into 24 h periods so that a direct comparison can be made.

p.11, lines 10-16: Why did the authors not compare the levels of the oxidation products instead of comparing the levels of their pre-cursors? The concentrations of the oxidation products of the cVMS have been calculated, and it should be possible to calculate them for SO2 as well. Given that one of the innovations of this work was the modeling of the cVMS oxidation products, I do not understand why these data are not used.

p.12, line 11: Has this been proven? It seems like a reasonable assumption, but I did not find proof for it in the paper.

Figures 1 and 2: The concentration intervals represented by the different colors are inconsistent. Could you not use a consistent logarithmic scale?

Figure 3: Information should be provided about the time period (dates) for which model data were collected and for which measured data were collected.

---

## Referee Comment (RC2) · Anonymous Referee #2 · 22 Feb 2017

The authors develop an emissions inventory of siloxanes, which they then model in CMAQ. They find that the spatial concentrations of parent and oxidized products are different. They also assess seasonality and vertical gradients of siloxanes. For the most part, the analysis is technically sound, although I do have some critiques. The manuscript is well composed, and advances understanding of the atmospheric impacts from personal care products. Overall, my recommendation is for publication of this paper with some revisions.

General Comments

1. As I understand from Page 5, the authors' construct their inventory using methods described by McLahlan et al. (2010), which are mainly based on antiperspirant sales.

Yet, the first sentence of the manuscript mentions that siloxanes are also present in sealers, cleaning products, and silicone products. To what extent are the authors' underestimating emissions by only considering antiperspirant sales? MacKay et al. (2015) suggest that antiperspirants only account for ∼70% of D5 consumption in personal care products in Canada. Buser et al. (2014) report per capita D5 emissions of 190 mg/person/d, yet the emissions used in this study are ∼30% lower. This is confusing, since Buser et al. also forms the basis of this study's emissions estimate (Page 5, Line 11). Table S2 (which I really liked, and think warrants inclusion in the main text rather than in supplemental), shows a wide range of per capita siloxane emissions. It is not clear that the authors' emissions are central estimates compared with the prior literature. Some text justifying the authors' selection of the emissions estimation technique would be helpful.

2. The authors' estimate D4 and D6 emissions by ratio to D5 from Chicago measurements (Page 5). However, there appears to be significant variability in in D4/D5 and D6/D5 emission ratios in the literature (Table S2). Tang et al. (2015) highlight that the emissions of D4 and D6 from personal care products are 1-2 orders of magnitude smaller than D5. Whereas the ratio used in this study for D6 is about an order of magnitude lower than for D5, and the ratio for D4 is only a factor of 3 lower. Again, it is not clear that the authors' emissions of D4 and D6 are central estimates compared with the prior literature, and some justification on why the authors' chose Chicago emission ratios would be helpful.

3. In the abstract, the authors' highlight that siloxanes have a high dependence on population density. I think it is important to mention that while this looks like the case for D5, it appears to be less so for D4 and D6. For D4 and D6, the model exaggerated urban-rural contrasts in Figure 3, and half the data points in Figure 4 were off by an order of magnitude compared to observations.

Specific Comments

[Figure]

Introduction

4. Page 1, Line 24. Where does the 4.5 x 10ˆ5 kg/y produced number come from? Citation needed.

5. Page 3, Lines 4-7. First mention of the DEHM and BETR models, and should be spelled out/described here, rather than later in the manuscript.

Methods

6. Page 4, Line 20. Is there a reference for the meteorology used that the authors' can cite? If not, what settings were used to generate the meteorological fields in WRF?

7. On Page 4, Line 25, kOH values are reported for parent molecules. However, it's not clear if oxidation products react away as well, and if this is taken into account.

8. Page 5, Lines 9-14. It is not clear why the authors' chose the methodology they did for estimating D4, D5, and D6 emissions, given the range of literature values shown in Table S2. Some justification here is needed.

9. Page 5, Line 25. What version of the NEI are 2004 emissions based on? Also, satellite trends of NO2 have shown significant decreases over the U.S. from 2005-2011 (Russell et al., 2012). By using an inventory that is 5-7 years out-of-date between the model and observations, how would modeling results be affected if NOx emissions were lowered (especially with respect to Figure 2)? Also, what biogenic emissions inventory (and version) is used?

Results

10. Page 7, Line 17. It is not clear why the authors' state that rural and remote locations follow an OH-induced seasonal pattern when the statistical relationships find only wind speed of significance (Page 8, Line 4). Suggest revising this statement here and in the conclusions section.

11. Section 3.4. Given the large discrepancies in model and observations for D4 and

D6 (see Comment 3), especially in rural locations (Figures 3 and 4), Figures 5 and 6 seem like a stretch. I suggest removing these figures and section, which can be done without any loss to the main findings of the manuscript.

Tables and Figures

12. I found Tables 2 and 3 hard to follow. The point on the relationship with population could be better made with a scatter plot with population on the x-axis, and D4/D5/D6 concentrations on the y-axis. Also, how is population determined for sites located in parks?

13. Figure 3. Would be helpful to include in the caption the year and region of the sampling, for readers unfamiliar with the Yucuis et al. study. Also, state abbreviations in the figure labels would be helpful.

14. Figure 4. The flow of this figure was confusing to me at first. It may help to put all the CMAQ vs. measured concentration plots in one column, and other model comparisons in the right column. Also, it would help to label the horizontal resolution of each model, for readers who may be unfamiliar with BETR and DEHM.

15. Figure 8. I think this figure could benefit from having the same x-axes. The vertical concentration gradients do not appear as sharp for o-D5 than for the parent molecule, which is instructive.

Minor Comments

16. Page 2, Line 17. References listed here look misplaced.

References

Mackay, D., and I. van Wesenbeeck (2014), Correlation of Chemical Evaporation Rate with Vapor Pressure, Environ Sci Technol, 48, 10259-10263, doi:10.1021/es5029074.

Russell, A. R., L. C. Valin, and R. C. Cohen (2012), Trends in OMI NO2 observations over the United States: effects of emission control technology and the economic recession, Atmos Chem Phys, 12, 12197-12209, doi:10.5194/acp-12-12197-2012.

---

## Author Comment (AC1) · 16 Apr 2017

**Response to Reviewers**

We thank both reviewers for their constructive comments and suggestions. Our responses to the reviewers' comments are indicated in blue italics. The revised manuscript follows.

**Reviewer 1**

General Comments

This is a strong paper. The methodology is of high quality, the discussion of the results is insightful, and the paper is very well written. Large scale modeling of the atmospheric fate of cVMS has been previously conducted by others using similar methods and the general principles governing atmospheric fate of these chemicals have been discussed. The major technical innovations in this paper are:

- A higher spatial resolution, which allows a better description of concentrations in large urban areas.

- The inclusion of oxidation products in the model, which allows the concentration fields of the products to be explored.

The major contributions of this paper to understanding are:

- A more thorough discussion of how cVMS emission patterns and properties affect the horizontal, vertical and temporal variability in concentrations of cVMS and their primary oxidation products. The horizontal and temporal variability for cVMS has been discussed earlier, but not in this depth. The vertical variability for cVMS is novel, as is the discussion of the cVMS oxidation products.

- A comparison of the model results is made with a larger and more diverse set of measurements than has been presented previously attempted, lending more confidence to the modeling, in particular for D4 and D6.

A disappointment with this paper is that more effort was not devoted to exploring the atmospheric fate of the cVMS oxidation products. Although these model data represent one of the major technical innovations of the work, the vast majority of the discussion is devoted to discussing the native cVMS.

*Response: The additional work on atmospheric fate of the oxidation product is work in progress. More detailed work on the oxidation fate requires semivolatile aerosol partitioning of the cVMS to be coded into the CMAQ model. We felt that the work was of sufficient contribution to the field to publish in two stages – one focusing on gas phase concentrations (the paper in question) and a followup paper that includes aerosol partitioning of the oxidation product. The aerosol yield information is just becoming available in the published literature now (in 2017) and would have been premature at the time of submission (in 2016).*

Specific Comments

Page 2, line 4: Here the authors take sides in an ongoing infected scientific/political debate. They support this position by citing papers representing just one side of the debate. This is unnecessary and does not contribute to the credibility of their work.

*Response: We have revised this section (p. 2, lines 2-14) by shortening it and referring the reader to a list of major regulatory screenings, reviews, and recent articles on the subject.*

Page 2, lines 16-17: Here several references are given for work that was published in just one paper, even though that one paper is expressly stated at the beginning of the sentence.

*Response: The additional citations, which contain measurement comparison to the DEHM or BETR models, have been removed. These papers are referenced later as examples of model-measurement comparisons.*

Page 2, lines 24-27: I am surprised by the selection of information presented here, given that the second author has published simulations that give much more detailed insight into the latitudinal gradients, urban-remote gradients, seasonal patterns and sensitivity to processes and parameterizations on a hemispheric scale.

*We have tried to modify this paragraph (p. 2, line 23 – p.3, line 8) to improve it along the lines of the reviewer comment.*

Page 3, line 4: McLachlan et al. (2010) had no measured data from summer and autumn. Incorrect citation.

*Response: We have separated citations (p. 3, lines 20-22) for McLachlan et al. (2010) and Krogseth et al. (2013) to remove confusion of which work corresponds to which period. We have changed our definition of the seasons to match those defined in the respective papers. The paper now states McLachlan et al. (2010) covering late spring and Krogseth et al. (2013) covering late summer.*

Page 4, line 26 and paragraph that follows: It is not clear how dry deposition was modeled. Were the surfaces considered to be infinite sinks? Is this a reasonable assumption for the cVMS? I am not aware of evidence showing that cVMS are rapidly degraded on surfaces, except on soil when it is very dry. cVMS have log octanol-air partition coefficients of about 4-6, which suggests that surface media have a limited ability to soak up cVMS and that a partitioning equilibrium between the atmosphere and surfaces would be reached quickly. In this case, assuming that surfaces are infinite sinks would be a poor assumption. While I doubt that this will modify the conclusions of this study, it is important that the assumptions made in modeling deposition are clearly stated and justified.

*Response: Cyclic siloxane dry deposition in CMAQ 4.7.1 is handled as an infinite sink. A deposition velocity is calculated using Pleim m3dry method that takes into account mixing and turbulence, molecular properties, and land type (Byun et al., 1999). The deposition velocity is multiplied by a concentration to calculate the amount deposited. This is consistent with how other species are treated in CMAQ. The Methods section (p. 5, lines 26-30) has been updated to reflect this.*

*A paragraph (p. 10, line 32 – p. 11, line 7) has been added to the Results and Discussion section describing that gas concentrations may be underpredicted due to the assumption that surfaces are treated as infinite sinks for deposition. We also explain that based on $logK_{OA}$ and $log\ K_{AW}$ values, we would expect the influence to be minimized for the parent compounds since deposition is a minor process. For the oxidized species, the influence would be expected to be larger but surface degradation of the oxidized species is unknown. We also found similar or lower $logK_{OA}$ values for other CMAQ deposition species.*

Page 6, line 31 and elsewhere: The literature cited by the authors discusses other sources of cVMS emissions besides personal care products. For instance, Brooke et al. (2009c) list emissions of residuals in PDMS as being twice as important as personal care products for emissions of D4 to air.

*Response: We had already acknowledged non-personal care product sources in the introduction but have rephrased the section in question (p. 8, lines 10-13) to add the details from the UK risk assessment (Brooke et al., 2009a, b, c). The Brooke et al. (2009 a,b,c) emission estimates for Europe indicate the major source of D4 to the air is from residual PDMS while D5 and D6 is from personal care product use.*

Page 8, lines 20-22: This assumption depends on the number of measurements that were conducted and the time period over which the measurements integrated. A handful of very short measurements can well give a non-representative picture of "typical" summer concentrations. I suggest that the authors provide some information on the number and length of the measurements.

*Response: Yucuis et al. (2013) measurement details have been added to paper (p. 10, lines 3-6).*
*   Chicago, IL: 16 measurements, 12 h, 8/13 – 8/21*
*   Cedar Rapids, IA: 4 measurements, 24 h, 6/29, 7/2, 7/14, 7/26*
*   West Branch, IA: 5 measurements, 30 – 47 h, 7/6, 7/15 – 7/22*
*A caveat (p. 10, lines 8-9) about the representativeness of the Yucuis et al. (2013) measurements has been added.*

p. 10, lines 18-20: Why are hourly modeled data compared with daily measurements when it is clear that the different time scales makes the comparison difficult? I suggest that you integrate the modeled data into 24 h periods so that a direct comparison can be made.

*Response: Thanks, we agree that it makes the most sense to average CMAQ data to 24-hour intervals. Table 2 and Section 3.3.3 of the text have been updated.*

p.11, lines 10-16: Why did the authors not compare the levels of the oxidation products instead of comparing the levels of their precursors? The concentrations of the oxidation products of the cVMS have been calculated, and it should be possible to calculate them for SO2 as well. Given that one of the innovations of this work was the modeling of the cVMS oxidation products, I do not understand why these data are not used.

*Response: In summary, we thank the reviewer for this excellent suggestion, but defer its implementation until future work for the reasons stated below. Furthermore, our rationale for looking at the precursor ratios is explained: because of the similar oxidation kinetics with OH, we believe that looking at precursor ratios is a useful conceptual model, particularly for the aerosol science community, which has given considerable investigation to the sulfur oxidation process. Furthermore, this conceptual model ties in nicely with experimental measurements of the precursor ratio (SO2/cVMS) and the aerosol ratio (S/Si), which for example was available in Bzdek et al. (2014).*

*We had not thought of examining the oxidation product concentration ratios, and we will explore that in our subsequent work that includes true aerosol partitioning of the oxidation products. It is complex to do (because of the different compartments that S(IV) and oxidized cVMS partition to). This complexity would require considerable additional word count in methods and results and discussion to accommodate. We anticipate oxidized cVMS to partition between gas phase, aerosol phase, and cloud droplets. Sulfur has the same phase partitioning issues, plus it can exist in different levels of dissociation (H2SO4, HSO4-, SO4--). So while such a ratio is likely possible, multiple ratios for different compartments would need to be considered – i.e., aerosol, cloud droplet, gas phase.*

p.12, line 11: Has this been proven? It seems like a reasonable assumption, but I did not find proof for it in the paper.

*Response: Thank you for the correction, as the statement has not been proven, but is rather an assumption. We have rephrased (p. 14, line 18) to state that we hypothesize increased model errors due to uncertain D4 and D6 emission rates.*

Figures 1 and 2: The concentration intervals represented by the different colors are inconsistent. Could you not use a consistent logarithmic scale?

*Response: Both figures now use a logarithmic scale, however the range is different from Figure 1 to 2 – so a given color does still indicate one concentration for cVMS and another for o-cVMS. We prefer to have separate scales in order to visually show concentration variability for each compound. While this may lead to confusion about relative concentrations, the fact that cVMS concentrations are much higher is noticeable throughout the paper, including in the abstract and in tables such as Table 1.*

Figure 3: Information should be provided about the time period (dates) for which model data were collected and for which measured data were collected.

*Response: The relevant details have been added to the now Figure 4 caption and the corresponding text.*

Methods

6. Page 4, Line 20. Is there a reference for the meteorology used that the authors' can cite? If not, what settings were used to generate the meteorological fields in WRF?

*Response: The following WRF details have been added to the methods section on page 5, lines 8-11:*

*"WRF was run with time steps of 120 s, 30 vertical layers, Morrison double-moment microphysics scheme, RRTMG longwave and shortwave physics scheme, Pleim-Xiu surface layer, Pleim-Xiu land surface model with two soil layers, and ACM2 PBL scheme. Reanalysis nudging using North American Regional Reanalysis (NARR) data was performed every three hours."*

7. On Page 4, Line 25, kOH values are reported for parent molecules. However, it's not clear if oxidation products react away as well, and if this is taken into account.

*Response: Additional details on the handling of the oxidation species has been added (p.5, lines 16-21) to explain that the oxidation products are believed to undergo additional oxidation reactions (Whelan et al. 2004) but they will remain oxidized cVMS – and thus are correctly labeled in the mechanism scheme and in the figures. Their physical properties will change upon further oxidation, but details on the mechanism, kinetics, and properties are very limited. In our treatment, the first oxidation step is modeled; subsequent oxidation steps are not modeled.*

*Added text:*

*"Reactions of the oxidation products are not included in the model. In part, this is because information is limited on the kinetics of further oxidation and on the changes that this would cause for fate, transport, and properties. Whelan et al. (2004) modeled subsequent oxidation reactions, and chamber-based oxidation studies observe multiple substitution products likely due to multiple substitution reactions or auto-oxidation by internal rearrangement (Wu and Johnston, 2016). In the model, only the first oxidation is computed. The oxidation products are denoted o-D4, o-D5, and o-D6, and for calculation of physical properties relevant to deposition, the single OH substitution is assumed."*

8. Page 5, Lines 9-14. It is not clear why the authors' chose the methodology they did for estimating D4, D5, and D6 emissions, given the range of literature values shown in Table S2. Some justification here is needed.

*Response: Please see the response to general comments one and two.*

9. Page 5, Line 25. What version of the NEI are 2004 emissions based on? Also, satellite trends of NO2 have shown significant decreases over the U.S. from 2005-2011 (Russell et al., 2012). By using an inventory that is 5-7 years out-of-date between the model and observations, how would modeling results be affected if NOx emissions were lowered (especially with respect to Figure 2)? Also, what biogenic emissions inventory (and version) is used?

*Response: Emissions were calculated from NEI 2002, version 3, with on-road and point sources projected to 2004 using EGAS, the EPA's Point source and Economic Growth Analysis System. Biogenic emissions were from BEIS 3.13. The emission model details have been to page 6, line 33 – page 7, line 2. The impact on the results of the dated emission inventory is likely only relevant for SO2:Si ratios, as SO2 emissions have dropped sharply. Ozone levels have decreased, but the cVMS and o-cVMS do not interact directly with ozone in our simulations. The influence of ozone, NOx, and VOC changes on OH concentrations (which will influence cVMS and o-cVMS) is difficult to anticipate in size, direction, and timing.*

Results

10. Page 7, Line 17. It is not clear why the authors' state that rural and remote locations follow an OH-induced seasonal pattern when the statistical relationships find only wind speed of significance (Page 8, Line 4). Suggest revising this statement here and in the conclusions section.

*Response: The general behavior of the rural sites is established in section 3.2, with description of the month of maximum and minimum concentration (see also Figure S6). Due to the normalization, the regression is really testing the ability of seasonal variability in local OH to predict seasonal variability in cVMS. In other words, do sites with the highest winter:summer ratio of D5 also have the highest winter:summer ratio of local OH$^{-1}$. The regression analysis says this is not the case. See Figure S8(B,D). The failure of the regression analysis to show predictive power of local OH ratios is different from the influence of regional OH on regional concentrations.*

*However, the regression analysis is important, from our perspective, because of the way that it shows the combination of PBL height and wind speed (or rather the season-to-season variation in this) is a predictor of season-to-season variability in cVMS concentrations. See Figure S7(A,C). The text has been modified (p. 9, lines 13-15) to stress that the regression analysis is testing correlation in season-to-season variability across seasons and sites.*

11. Section 3.4. Given the large discrepancies in model and observations for D4 and D6 (see Comment 3), especially in rural locations (Figures 3 and 4), Figures 5 and 6 seem like a stretch. I suggest removing these figures and section, which can be done without any loss to the main findings of the manuscript.

*Response: We would like to leave these figures in the paper. The assumptions behind them are well documented, and we have changed the text to stress that the figures in question are tied to the assumption of population density dependent emissions of D4, D5, and D6 (p. 13, lines 4-6). We feel that figures like this may help measurement scientists construct interesting sampling plans, and to establish preliminary sample times and air volumes necessary to get appropriate sample loadings. The fact that a model with population-dependent emissions and realistic spatio-temporal OH predicts these types of ratios is a valuable result. Future measurements may prove it wrong – and more D4 and D6 measurements and an emission inventory are needed.*

Tables and Figures

12. I found Tables 2 and 3 hard to follow. The point on the relationship with population could be better made with a scatter plot with population on the x-axis, and D4/D5/D6 concentrations on the y-axis. Also, how is population determined for sites located in parks?

*Response: Thank you for the excellent figure suggestion. We have moved the tables to the supplemental section (now Tables S4 and S5), as we feel they may be useful references for monitoring study design. Population was determined from gridded population estimates detailed in the supplemental section. The population is that within the 36 km grid cell. For large metro areas that span multiple grid cells, the grid cell with highest population was used for analysis.*

*Population data was from population surrogates downloaded from EPA 2011 v6.0 Air Emissions Modeling Platform. This data is derived from census data, and therefore represents permanent population, and does not reflect seasonal visitors to tourist destinations. These details have been added to the paper on page 6, lines 31-33, and page 11, line 24.*

*Added following figure, now Figure 3:*

[Figure]

13. Figure 3. Would be helpful to include in the caption the year and region of the sampling, for readers unfamiliar with the Yucuis et al. study. Also, state abbreviations in the figure labels would be helpful.

*Response: Figure 4 has been updated with state labels and measurement details in the caption.*

14. Figure 4. The flow of this figure was confusing to me at first. It may help to put all the CMAQ vs. measured concentration plots in one column, and other model comparisons in the

right column. Also, it would help to label the horizontal resolution of each model, for readers who may be unfamiliar with BETR and DEHM.

*Response: Figure 5 layout has been updated along with resolution details to figure caption.*

15. Figure 8. I think this figure could benefit from having the same x-axes. The vertical concentration gradients do not appear as sharp for o-D5 than for the parent molecule, which is instructive.

*Response: Figure 9 has been edited to have consistent x-axes.*

Minor Comments

16. Page 2, Line 17. References listed here look misplaced.

*Response: Citations have been removed.*

*References*

[revised manuscript text omitted]

**Figure 4̶3̶:** Model comparison to Yucuis et al. (2013) ̶M̶i̶d̶w̶e̶s̶t̶ ̶m̶e̶a̶s̶u̶r̶e̶m̶e̶n̶t̶s̶. Model results ̶a̶r̶e̶ from CMAQ ̶m̶o̶d̶e̶l̶e̶d̶ ̶(̶July 1-30 simulation); ̶m̶o̶n̶t̶h̶ ̶(̶d̶a̶y̶s̶ ̶1̶-̶3̶0̶)̶ ̶w̶h̶i̶l̶e̶ measurements were conducted in 2011 from Aug 13-21 (Chicago), Jun 29-Jul 26 (Cedar Rapids), and ̶d̶a̶t̶e̶s̶ ̶w̶e̶r̶e̶ ̶i̶n̶ ̶2̶0̶1̶1̶ ̶f̶r̶o̶m̶ Jul 6-22 (West Branch), respectively. ̶f̶o̶r̶ ̶C̶h̶i̶c̶a̶g̶o̶,̶ ̶J̶u̶n̶e̶ ̶2̶9̶ ̶—̶ ̶J̶u̶l̶y̶ ̶2̶6̶ ̶f̶o̶r̶ ̶C̶e̶d̶a̶r̶ ̶R̶a̶p̶i̶d̶s̶,̶ ̶a̶n̶d̶
5   ̶J̶u̶l̶y̶ ̶6̶ ̶—̶ ̶2̶2̶ ̶f̶o̶r̶ ̶W̶e̶s̶t̶ ̶B̶r̶a̶n̶c̶h̶.̶ Hourly model data was averaged to 12, 24, and 36 hour periods, starting at typical measurement start

[revised manuscript text omitted]
$_4$ Frac. Bias | CMAQ D$_4$ Frac. Error | CMAQ D$_4$ Error | CMAQ D$_5$ Frac. Bias | CMAQ D$_5$ Frac. Error | CMAQ D$_5$ Error | CMAQ D$_6$ Frac. Bias | CMAQ D$_6$ Frac. Error | CMAQ D$_6$ Error | BETR D$_5$ Frac. Bias | BETR D$_5$ Frac. Error | BETR D$_5$ Error | DEHM D$_5$ Frac. Bias | DEHM D$_5$ Frac. Error | DEHM D$_5$ Error |
|---|---|---|---|---|---|---|---|---|---|---|---|---|---|---|---|
| Bratt's Lake, SK | -0.145 | 0.145 | 0.352 | 1.24 | 1.24 | 6.25 | 0.114 | 0.114 | 0.0376 | 0.383 | 0.383 | 0.900 | -0.303 | 0.303 | 0.500 |
| Whistler, BC | -1.89 | 1.89 | 43.7 | -0.354 | 0.354 | 1.93 | -1.56 | 1.56 | 1.31 | -1.01 | 1.01 | 4.30 | -0.485 | 0.485 | 2.50 |
| Downsview, ON | 0.591 | 0.591 | 9.23 | 0.390 | 0.390 | 26.6 | -0.518 | 0.518 | 2.55 | -1.58 | 1.58 | 48.5 | -0.651 | 0.651 | 27.0 |
| Fraserdale, ON | -1.45 | 1.45 | 4.53 | 0.0164 | 0.0164 | 0.0313 | -1.52 | 1.52 | 0.354 | 0.417 | 0.417 | 1.00 | 0.100 | 0.100 | 0.200 |
| Ucluelet, BC | -1.93 | 1.93 | 43.2 | -0.992 | 0.992 | 4.84 | -1.71 | 1.71 | 1.11 | - | - | - | -1.07 | 1.07 | 5.10 |
| Point Reyes, CA | -0.0195 | 0.0195 | 0.0813 | 0.848 | 0.848 | 9.56 | 0.215 | 0.215 | 0.137 | -1.38 | 1.38 | 5.30 | 0.667 | 0.667 | 6.50 |
| Sydney, FL | 0.613 | 0.613 | 4.77 | -0.674 | 0.674 | 41.3 | -0.753 | 0.753 | 2.19 | -1.71 | 1.71 | 75.5 | -1.38 | 1.38 | 67.0 |
| Groton, CT | 0.969 | 0.969 | 7.32 | -0.745 | 0.745 | 52.1 | -1.45 | 1.45 | 10.1 | - | - | - | -1.15 | 1.15 | 70.0 |
| Mean | -0.407 | 0.950 | 14.1 | -0.0334 | 0.658 | 17.8 | -0.897 | 0.980 | 2.22 | -0.812 | 1.08 | 22.6 | -0.534 | 0.726 | 22.4 |
| Median | -0.0825 | 0.791 | 6.05 | -0.169 | 0.709 | 7.91 | -1.10 | 1.10 | 1.21 | -1.19 | 1.19 | 4.80 | -0.568 | 0.659 | 5.80 |

$$Frac. Bias = \left( \frac{m - o}{\frac{m + o}{2}} \right)$$

$$Frac. Error = \left( \frac{|m - o|}{\frac{m + o}{2}} \right)$$

$$Error = |m - o|$$

**Section S11:  Vertical Concentrations**

[Figure]

**Figure S9**:  Grid cell locations for vertical analysis in Los Angeles area.

[revised manuscript text omitted]